# The Effect of an Electronic Passive Prompt Intervention on Prolonged Occupational Sitting and Light-Intensity Physical Activity in Desk-Based Adults Working from Home during COVID-19 in Ireland

**DOI:** 10.3390/ijerph20136294

**Published:** 2023-07-04

**Authors:** Aidan J. Buffey, Gráinne Hayes, Brian P. Carson, Alan E. Donnelly

**Affiliations:** 1Department of Physical Education and Sport Science, Faculty of Education and Health Sciences, University of Limerick, V94 T9PX Limerick, Ireland; grainne.hayes@ul.ie (G.H.); brian.carson@ul.ie (B.P.C.); alan.donnelly@ul.ie (A.E.D.); 2Physical Activity for Health Research Cluster, Health Research Institute (HRI), University of Limerick, V94 T9PX Limerick, Ireland

**Keywords:** workplace, occupational physical behaviours, sedentary behaviour, light-intensity physical activity, sedentary breaks, prolonged sitting, accelerometry

## Abstract

This study aimed to assess the effect of passive prompts on occupational physical behaviours (PBs) and bouts of prolonged sitting among desk-based workers in Ireland who were working from home during the COVID-19 pandemic. Electronic passive prompts were delivered every 45 min, asking participants to walk for five minutes, during working hours. Twenty-eight participants (aged 30–67 years) completed the six-week intervention between October 2020 and April 2021. PBs were measured using an activPAL3^TM^ accelerometer, following a 24 h wear protocol, worn for the duration of the study. Participants were highly sedentary at both baseline (77.71% of work hours) and during the intervention (75.81% of work hours). However, the number of prolonged occupational sedentary bouts > 90 min was reduced compared to baseline (0.56 ± 0.08 vs. 0.77 ± 0.11, *p* = 0.009). Similar reductions were observed in the time spent in sustained sitting > 60 and >90 min when compared to baseline sedentary patterns (60 min: −31.27 ± 11.91 min, *p* = 0.014; 90 min: −27.97 ± 9.39 min, *p* = 0.006). Light-intensity physical activity (LIPA) significantly increased during the intervention (+14.29%, *p* = 0.001). This study demonstrates that passive prompts, delivered remotely, can both reduce the number and overall time spent in prolonged bouts of occupational sedentary behaviour and increase occupational LIPA.

## 1. Introduction

Developments in technology have led to a significant reduction in the requirement for physical activity (PA) due to the automation and mechanisation of society [1,2], leading to an increase in desk-based occupations. Many individuals with desk-based occupations remain sedentary for prolonged periods of time, owing to the nature of their work [3]. Desk-based occupation tasks can be classified as sedentary behaviour, which is defined as any waking behaviour characterised by an energy expenditure ≤ 1.5 metabolic equivalents (METs), while in a sitting, reclining, or lying posture [4,5]. Consequently, individuals with desk-based occupations have been shown to accumulate high levels of sedentary behaviour during their occupational work hours. For example, Gilson et al. [6] found that two groups of office workers spent 68% and 74% of their work hours sedentary and in prolonged sedentary bouts (SBs), where the average longest bout was 100 and 111 min per day, respectively [6]. Similarly, Clemes et al. [7] found office workers spent 71% of their working hours sedentary.

The term, physical behaviours (PBs), encompasses PA, sedentary behaviour and sleep. Recent device-based literature examining PBs when working from home (WFH) during COVID-19 has emerged from Brazil [8] and Sweden [9]. Hallman et al. [9] examined the difference in the PBs of office workers when working at the office compared to when WFH during COVID-19. The findings illustrated that during occupational hours, participants were sedentary for 72.85% when working at the office compared to 72.48% of their occupational hours when WFH [9]. Whereas Brusaca et al. [8] did not distinguish domains such as occupational hours but showed that on workdays during COVID-19 restrictions, participants were sedentary for 46.5% of their total workday, compared to 47.9% of their workday pre-COVID-19 [8]. These studies indicate that the prevalence of sedentary behaviour is high when WFH and not significantly different when compared to working in the office both during COVID-19 [9] and pre-COVID-19 [8]. The lack of difference between the two environments may be explained by the Theory of Planned Behaviour (TPB) [10], to understand the high accumulation of occupational sedentary behaviour. The TPB postulates that subjective norms, attitude, and perceived behavioural control serve as the underlying determinants of a behaviour (sedentary behaviour) and predict intention and subsequent behaviour [11,12]. Niven and Hu [13] investigated office worker beliefs around reducing occupational sitting and accumulating 2 h of light-intensity PA (LIPA) and standing per day. Participants reported perceived difficulty in reducing sitting with reasons, such as reduced productivity, disrupted work and concentration, disapproval from work colleagues and managers, heavy demands of work, and the nature of the desk/computer-based role [13]. However, participants did state that changing external factors such as the physical work environment and workplace culture to increase the acceptability of increased standing and moving was required to facilitate change [13].

Workplace interventions designed to break up and reduce prolonged periods of sedentary behaviour in office workers are required [7]. Much of the recent literature focusing on interrupting prolonged sitting in the workplace has incorporated environmental changes and/or active workstations such as treadmill desks, cycling workstations, and sit-to-stand adjustable desks. However, these are associated with high costs, significant changes to the office’s physical environment and the employee’s typical work routines [14], and often rely on prompts to engage with the workstations. Developments in technology now allow computer software to trigger electronic screen-based passive prompts providing a message to encourage employees to break their bouts of prolonged sitting [15]. Therefore, passive prompts could be implemented to interrupt prolonged occupational sitting [16] and promote walking as a sedentary break, thereby increasing PA, the number of sedentary breaks, and replacing sedentary behaviour with PA as a low-cost intervention [17].

### Research Question and Aim

This study will address the lack of literature surrounding desk-based adult PBs and sedentary behaviour patterns when WFH in Ireland. As the level of restrictions varied across countries, the PBs may differ from those reported in Brazil [8] and Sweden [9]. Though, given the evidence that prolonged occupational sedentary behaviour when WFH during COVID-19 restrictions is comparable to working in an office during COVID-19 [9] and pre-COVID-19 [8], interventions that target reducing and interrupting occupational sedentary time are warranted in this emerging population of adults WFH. Furthermore, growing literature has begun to suggest that the pattern in which we accumulate total sedentary time has implications for an individual’s cardiometabolic health [18], with sustained uninterrupted prolonged sedentary time recognised as a distinct health risk [19]. However, frequent short physical activity breaks of LIPA have been shown to significantly improve cardiometabolic health markers such as postprandial glucose and insulin in acute, laboratory-based studies [20]. Therefore, this study aimed to (i) establish PB patterns of desk-based adults while WFH during COVID-19 restrictions in Ireland; (ii) investigate the effects of a remote passive prompt intervention with desk-based individuals who were WFH on occupational PBs and the pattern of sedentary behaviour accumulation. The passive prompts were intended to break bouts of prolonged occupational sitting, which were defined as >60 and >90 min, by passively prompting a sedentary break every 45 min with five minutes of light walking. The nature of the research question led to three *a priori-*developed hypotheses that this study aimed to investigate and test. Firstly, it was hypothesised that the intervention would increase occupational LIPA. Secondly, it was hypothesised that the intervention would reduce the number of sedentary bouts > 60 and the time spent in sustained sitting > 60 min. Finally, it was hypothesised that the intervention would reduce the number of sedentary bouts > 90 and the time spent in sustained sitting > 90 min.

## 2. Materials and Methods

### 2.1. Study Design and Procedure

This study was approved by the University of Limerick Faculty of Education and Health Sciences Research Ethics Committee (2020_06_25_EHS) and adhered to the Declaration of Helsinki. All participants provided their informed consent via a Qualtrics survey in which they selected that they consented to participation.

Data for this remote six-week AB within-subject quasi-experimental study were collected between October 2020 and April 2021. An AB design is a two-phase design composed of a baseline phase (A) and an intervention phase (B). The baseline phase (A) measured habitual 24 h and occupational PBs for three weeks, split into weeks one, two, and three. The intervention phase (B) investigated the impact of a passive electronic prompt intervention on 24 h and occupational PBs for three weeks, split into weeks four, five, and six (See Figure 1). The advantage of a within-subject design is that each participant serves as their own control, thereby controlling for individual differences. However, it is important to note the limitations of this design such as time-related effects such as the season, which could have influenced the outcomes and the reduced likelihood of assessing a cause-and-effect relationship.

### 2.2. Study Participants

This study used self-selection sampling where the study was advertised and interested individuals who fit the inclusion and exclusion criteria volunteered of their own accord. Individuals who were WFH in the Republic of Ireland during the COVID-19 pandemic restrictions were invited to participate through mass emailing to staff and affiliates of the University of Limerick via an email distribution list (>1000 email addresses) and through a public recruitment call for participants on Twitter. Individuals interested in participating in the research study were directed to an online survey (Qualtrics). This online survey presented the study information and participants provided their informed consent. Once participants provided their informed consent, they were asked to provide descriptive information and their contact details and postal address to arrange their enrolment in the study. The online survey responses were screened based on the study’s inclusion and exclusion criteria to ascertain if an interested individual was eligible to participate.

Participants that were aged between 30 and 67 years, who were desk-based, WFH, and capable of completing five minutes of light-intensity walking were eligible to participate. Participants were excluded upon survey review if the interested individuals had any health complication that would prevent them from performing five minutes of walking and if they were not following a structured work schedule, working part-time, or working in their workplace office. Individuals who indicated that they were working “hybrid” were eligible, where the vast majority of their work was completed at home with occasional visits to the workplace.

Participants were not screened for any clinical conditions such as cardiovascular, orthopaedic, or neurological diseases/disorders, due to the remote conduct of this study and the inclusion and exclusion criteria. However, any interested individual who indicated that they had a health complication, which meant that they were not able to complete five minutes of light-intensity walking was excluded from the study.

Demographic information and descriptive characteristics such as age, sex, height, and weight were self-reported via the Qualtrics survey.

### 2.3. Passive Prompt Intervention

The passive prompts were delivered by a desktop-based software program (BreakTimer, Open Source, Tom Watson). The software program allowed the research team to schedule passive prompts during the participants’ working hours, remotely. The software would firstly remind participants that they had been working for 45 min; this prompt was issued with a small on-screen pop-up, in the right corner of their screen. The pop-up notified participants that a break was about to begin, the software would then enlarge to the desktop’s full screen. At full screen, the software presented the following two phrases (1) “Time for a break!” and (2) “Please complete five minutes of light walking around your room/house/garden/street”.

Participants installed the desktop-based software program BreakTimer onto all their workplace devices prior to Week 4, and the commencement of the intervention phase (B). Participants were instructed to set up the BreakTimer software program and enable the program to run during their typical working hours. Participants followed the passive prompt intervention during their working hours from Week 4 through to the end of Week 6 (See Figure 1).

### 2.4. Physical Behaviours (Physical Activity, Sedentary Behaviour, and Time in Bed) Measurement

PBs at baseline and during the intervention were assessed via an activPAL3 Micro triaxial accelerometer (activPAL3) (PAL Technologies LTD., Glasgow, Scotland).

The activPAL3 along with a participant handbook was posted and participants were instructed to fit the activPAL3 upon its arrival. The device was initialised by the lead researcher (A.J.B.) and set to begin recording two days after being posted. In addition, a video and participant handbook were sent to the participants, via email, demonstrating how to fit the activPAL3 (Found here: https://www.youtube.com/watch?v=DxvcqLmK8M4 (accessed on 28 July 2020).

Participants were required to affix the activPAL3 to the anterior aspect of the midline of the right thigh, using a nitrile sleeve and waterproof Tegaderm dressing, in accordance with the manufacturer’s guidelines (Figure 2).

Over the six-week measurement period, participants followed a 24 h/day wear protocol and wore the activPAL3 for six consecutive days each week (Monday 5 a.m. to Sunday 5 a.m.). Participants were asked to remove the monitor only if they were going to be submerged in water for a prolonged period (i.e., bathing and swimming). Participants completed a weekly logbook, where they would note their daily work start and finish time, if the activPAL3 was removed, the reason for removal, and the time the device was removed and reaffixed. Participant’s logbooks were used to differentiate between activity domains i.e., occupational, leisure, and time in bed.

On Sundays, participants were instructed to remove the activPAL3 and connect it to a PC via a USB interface. The participants were provided with a link to download PALtransfer, a proprietary software. PALtransfer enabled participants to upload their activPAL data file to the PALportal (an online cloud storage facility), charge the activPAL3, and begin a new recording for the week ahead. Participants were instructed to refit the activPAL3 before going to sleep on Sunday evening.

### 2.5. Primary Outcomes

The primary outcome was change in occupational LIPA. Secondary outcome measures relating to our research question were change in the number of prolonged occupational SBs both >60 and >90 min and time spent in sustained occupational SB > 60 and >90 min. To address the research question and secondary outcomes, the number of SBs > 40 min and the time spent in sustained occupational SBs > 40 min were also assessed.

### 2.6. Data Extraction and Processing

Participant’s activPAL data files were downloaded remotely from PALportal, by the lead researcher (A.J.B.). Once downloaded, the activPAL data files were processed using the proprietary activPAL software (PALanalysis v8.11.5.64) and exported as 15 s epoch .csv files, which are compatible with Microsoft Excel [21].

To be included in this study, participants were required to provide data for at least one valid working day in each week of the six-week study. A valid working day was defined as a 24 h recording from 5 a.m. to 5 a.m., with ≤4 hour’s non-wear time. Non-wear time followed previous definitions as a period of ≥60 min of consecutive zero activity counts [22,23,24].

The exported 15 s epoch .csv files were processed in Microsoft Excel as reported by Dowd et al. [23]. The raw Excel spreadsheet reported the number of seconds that participants were engaging in sedentary, upright, stepping, primary lying, secondary lying, non-wear time behaviour, and the number of sedentary to upright movements and upright to sedentary movements.

The first step in processing the exported 15 s epoch .csv file was to calculate the sum of the vector magnitudes (SVM), which was calculated by summing the value from the three orthogonal axes. The SVM was selected in place of the vertical axis only as the activPAL3 is a tri-axial accelerometer and, thereby, using the SVM has been suggested to provide greater sensitivity when determining LIPA [25].

The SVM was calculated as a new column named “SVM” on the raw 15 s epoch data files, which summed the exported raw data per 15 s epoch row named “Sum(abs(dChannel1))”, “Sum(abs(dChannel2))”, and “Sum(abs(dChannel3))” (See Equation (1) [25,26]).

Equation (1) Displaying the Formula used to calculate SVM.
√(X^2^ + Y^2^ + Z^2^) (1)

MVPA (>3 METS) was calculated and defined as an SVM cut point > 8873. This MVPA (>3 MET) > 8873 counts.15 s^−1^ cut-point was previously validated in a population of adults aged 39.9 ± 11.5 using the COSMED K4b^2^ (COSMED, Rome, Italy), a breath-by-breath portable metabolic unit [25].

To calculate the time spent at >8873 in the 15 s epoch, the raw “stepping time” output was used. The 15 s epoch row was read and if the row showed an SVM > 8873, the time spent stepping was reported.

LIPA was calculated by subtracting sedentary time, primary lying time, secondary lying time (directly from the acitvPAL output), standing time, and MPVA time from the 15 s epoch for each row in the data file. Any time remaining in each 15 s epoch was therefore identified as LIPA.

The activPAL proprietary algorithms and subsequent software accurately estimate lying, sitting, and standing, and, therefore, the raw output of lying, sitting, and standing were used in place of a sedentary (≤1.5 MET) cut-point [25]. Standing was calculated by subtracting stepping time from upright time in the raw activPAL 15 s epoch .csv file.

Sedentary time over the 24 h period was calculated by summing the raw activPAL 15 s epoch .csv output of “sedentary”; “primary lying”, and “secondary lying” from 5 a.m. to 5 a.m. and repeated for each day of the recorded week. To calculate sedentary time during occupational hours, the participants’ start of work and end of work, as self-reported in their weekly logbook, were used to identify the corresponding epochs. Subsequently, the rows between the start and end of work epochs were summed to calculate occupational sedentary time and this was repeated for each workday of the recorded week.

This two-step process was repeated for standing, stepping, LIPA, and MVPA, where each column was summed as documented above to provide both the 24 h PBs (5 a.m. to 5 a.m.) and occupational PBs (self-reported work hours); this allowed the total time spent in each PB to be calculated for the participants’ occupational work hours and over the 24 h period.

Participants’ daily and occupational PBs, for valid days, i.e., where the participant worked (i.e., Monday to Friday), were then averaged to calculate the mean for each week of monitoring.

### 2.7. Sedentary Bouts and Time Spent in Sustained Sedentary Behaviour Processing

The sedentary pattern, i.e., number of SBs of different bout lengths and the time accrued in these different bout lengths (>40, >60, and >90 min) were examined by a MATLAB (The MathWorks Inc., Natick, MA, USA) custom software program. The MATLAB custom software program has been described in detail previously by the creator [23]. Briefly, the MATLAB software program read imported 15 s epoch .csv raw activPAL data files and binary coded each epoch. A sedentary epoch was defined as an epoch spent in entirely sedentary behaviour (code = 1) whereas a non-sedentary epoch was defined as an epoch with <15 s in sedentary behaviour (code = 0). To examine the pattern of sedentary behaviour, the program identified one sedentary epoch as the beginning of an SB and the last consecutive sedentary epoch as the end of the SB. The MATLAB program sequentially examined each recorded day in the raw activPAL 15 s epoch file and provided the start and end time for each continuous sedentary period into bouts and provided the minutes and seconds. The sedentary patterns and SB information were then written into a new Microsoft Excel file.

The new Microsoft Excel sheet, from the MATLAB output, with the SB information was processed and the total number of SB and time spent sustained in the SB > 40 min, >60 min, and >90 min was calculated. Furthermore, the daily total number of SBs and total time spent sedentary were reported. Similar to PBs, the sedentary behaviour patterns (both number and time in sustained SB; >40, >60, and >90 min) were processed and analysed over the 24 h period (Appendix A), during occupational hours (Table 3) and during leisure time (Appendix A).

Time in bed, a proxy measure for sleep, was estimated using the last registered non-sedentary epoch followed by a continuous sedentary period (>2 h), which was identified as the time participants went to bed, whilst the first non-sedentary epoch of the following day was identified as the rise time [23]. Time in bed was calculated as (time between 5 a.m. and the first non-sedentary epoch) + (time between the last non-sedentary epoch and the next 5 a.m. time point) [23]. The measure of time in bed was subjective as the participants did not self-report their sleep time and, therefore, the decision of time to bed and rise time was made by the lead researcher (A.J.B.) based on the above criteria and available data, from the MATLAB output. Breaks during time in bed (non-sedentary epochs; i.e., bathroom breaks) were identified by manual visual assessment of the data. This time in bed was subtracted from the 24 h total measurement to provide waking time.

Due to the nature of the study design and the data processing approach, this study presents the participants’ PBs during occupational hours (Table 2) and over the 24 h period (Table 4). This study presents occupational hours first as the intervention took place only during working hours and the primary research questions were related to changes in occupational sitting and LIPA. Some participants were working “hybrid” where they would work some days from their university office and others WFH; however, the days working at the office were minimal due to our inclusion and exclusion criteria and the COVID-19 restrictions.

### 2.8. Statistical Analysis

Descriptive statistics of the participants with valid data and who were included in the analysis are reported in Table 1. Statistical analyses were performed using SPSS Version 28 (IBM Corp., Armonk, NY, USA).

The data were checked for parametricity and normal distribution before conducting a 2 (Condition: Baseline/Intervention) × 3 (Time: Week 1/Week 2/Week 3) repeated measures ANOVA. Whereby the condition is calculated as the mean of the three weeks, respectively (Baseline; Week 1, Week 2, and Week 3; Intervention; Week 4, Week 5, and Week 6) and Time was reported as the mean of the respective week at the three time points for the condition, i.e., Week 1 vs. Week 4, Week 2 vs. Week 5, and Week 3 vs. Week 6.

The 2 × 3 repeated measures ANOVA was used for confirmatory purposes, whereby our *a priori* developed hypotheses were tested, rendering it unnecessary to adjust the level of *α* [27]. By performing confirmatory hypothesis testing rather than exploratory hypothesis formation, the risk of multiple-comparison is greatly mitigated as the authors have stated that the interest lies in the main effect of the first factor (condition) for the primary and secondary outcome variables [27].

Mauchly’s test of sphericity was used to test if the assumption of sphericity was met for the repeated measures ANOVA. Where the assumption of sphericity was violated (*p* < 0.05), a correction was applied to produce a valid *F*-ratio, which involved adjusting the degrees of freedom associated with the *F-*value. Specifically, if the assumption of sphericity was violated and when >0.75 the Huynh–Feldt correction was applied and when <0.75 then the Greenhouse–Geisser correction was applied.

Where a main effect was reported for Time, a pairwise comparison was conducted with Bonferroni correction.

Cohen’s *d* was reported as a measure of effect size and calculated to quantify the magnitude of difference in change from baseline to intervention per standard guidelines; effect sizes of 0.2, 0.5, and 0.8 were considered small, moderate, and large, respectively [28] and *α* was set at 0.05.

An *a priori* formal sample size calculation was not performed.

## 3. Results

### 3.1. Participant Flow

In total, 60 individuals expressed interest in participating in this study. Of the 60 individuals, 57 met the inclusion criteria and 45 agreed to complete the intervention and wear the accelerometer. Of the 45 that agreed to take part, 28 provided valid data for the six-week measurement period. Of the 28 participants with valid data, 18 (64.29%) were WFH for the full duration of the study and 10 (35.71%) participants reported attending the office on at least one day during the study.

### 3.2. Descriptive Characteristics of the Included Participants:

Descriptive statistics for the 28 participants can be observed in Table 1.
ijerph-20-06294-t001_Table 1Table 1Participant characteristics presented as Mean (±Standard Deviation).
TotalMaleFemale*N*281018Age (Years)46.8 (±8.3)48.3 (±6.8)45.9 (±9.1)Stature (Cm)172.28 (±9.14)181.50 (±5.55)167.16 (±6.28)Body Mass (Kg)78.85 (±17.40)87.21 (±8.03)74.20 (±19.36)BMI (kg/cm^2^)26.57 (±5.88)26.51 (±2.61)26.60 (±7.07)Abbreviations: BMI; Body Mass Index.


### 3.3. Occupational Light-Intensity Physical Activity When Working from Home

The two-way repeated-measure ANOVA test indicated that there was a significant main effect of the intervention on the average amount of LIPA completed during working hours (*F* (1,27) = 13.015, *p* = 0.001). The participant’s habitual LIPA time was 0.35 ± 0.02 h at baseline compared to 0.4 ± 0.03 h, during the intervention, an increase of 14.29% (Table 2).

Mauchly’s test indicated that the assumption of sphericity had been violated when comparing the three time points measured for LIPA during working hours (X^2^ (2) = 12.06, *p* = 0.002). Applying the Huynh–Feldt correction, there was no main effect of time on LIPA (*F* (1.52, 41.04) = 0.024, *p* = 0.95).

There was no interaction between the intervention and time (*F* (2, 54) = 0.39, *p* = 0.679).

### 3.4. Total Occupational Sitting When Working from Home

At baseline, when WFH, desk-based employees worked on average for 8.39 ± 0.15 h, of which participants were sedentary for 6.52 ± 0.23 h. During the intervention period, the participants’ average work hours were 8.31 ± 0.17 h, of which participants were sedentary for 6.3 ± 0.21 h (Table 2). The results of the two-way repeated-measures ANOVA showed that there was no main effect of the intervention for work hours (*F* (1, 27) = 0.271, *p* = 0.607) or occupational sedentary time (*F* (1, 27) = 1.904, *p* = 0.179), respectively.

There was no main effect of time on work hours (*F* (2, 27) = 0.739, *p* = 0.482) or occupational sedentary time (*F* (2,27) = 1.603, *p* = 0.211).

There was no interaction between the intervention and time on work hours (*F* (2,27) = 0.876, *p* = 0.422) or sedentary time (*F* (2, 27) = 1.354, *p* = 0.267).
ijerph-20-06294-t002_Table 2Table 2This table presents the PBs during self-reported occupational hours throughout the baseline (Week 1–3) and intervention (Week 4–6). Presented as Mean ± Standard Error (95% Confidence Intervals).Occupational Physical BehavioursBaselineInterventionMean Change*p**d*Work (h)8.39 ± 0.15 8.31 ± 0.17 −0.08 ± 0.15 0.6070.01(8.08–8.70)(7.96–8.66)(−0.45–−0.07)Sedentary Time (h)6.52 ± 0.23 6.3 ± 0.21 −0.23 ± 0.16 0.1790.07(6.05–7.00)(5.88–6.72)(−0.56–0.11)Standing Time (h)1.39 ± 0.11 1.45 ± 0.13 +0.06 ± 0.07 0.390.03(1.16–1.61)(1.19–1.71)(−0.08–0.21)*** Stepping Time (h)****0.54 ± 0.05****0.61 ± 0.05****+0.06 ± 0.03****0.023****0.18****(0.45–0.63)****(0.51–0.7)****(0.01–0.12)**Non-Wear Time (h)000--**LIPA (h)****0.35 ± 0.02****0.40 ± 0.03****+0.05 ± 0.01****0.001****0.33****(0.30–0.40)****(0.34–0.45)****(0.02–0.07)**MVPA (h)0.20 ± 0.03 0.21 ± 0.03 +0.01 ± 0.02 0.6470.01(0.14–0.25)(0.14–0.27)(−0.03–0.05)MVPA (Mins)11.58 ± 1.64 12.31 ± 1.79 +0.73 ± 1.15 0.5280.02(8.21–14.95)(8.64–15.99)(−1.62–3.08)Steps2744.22 ± 262.29 2934.96 ± 270.50 +190.74 ± 169.26 0.2700.05(2206.06–3282.39)(2379.95–3489.97)(−156.56–538.04)Abbreviations: h; hours. Mins; minutes. LIPA; light-intensity physical activity. MVPA; moderate-to-vigorous physical activity. **Bold** indicates that a significant change was found for the reported outcome variable (*p* < 0.05). Time is reported as hours, with minutes expressed as decimals 0–100, whereby any reader interested in converting hours to minutes can multiply the reported outcome variable by 60; for example (1.45 h × 60 = 87 min). * Stepping: The stepping reported is the total accumulated stepping time, where the intensity of the stepping is then further classified into LIPA and MVPA time.


### 3.5. Occupational Sedentary Bouts When Working from Home

The mean (±SD) number of SBs > 60 min during working hours, in the baseline phase, was 1.64 ± 0.16 compared to 1.38 ± 0.16 during the intervention phase, this difference did not reach significance (*F* (1, 27) = 4.108, *p* = 0.053). However, the number of sedentary bouts > 90 min was significantly (*F* (1, 27) = 7.978, *p* = 0.009) less during the intervention (0.56 ± 0.08) compared to the participants’ habitual behaviour (0.77 ± 0.11) (Table 3).

There was no main effect of time on sedentary bouts > 60 min during working hours (*F* (2, 54) = 1.259, *p* = 0.292) or sedentary bouts > 90 min during working hours (*F* (2, 54) = 1.158, *p* = 0.322).

Mauchly’s test indicated that the assumption of sphericity had been violated when comparing the interaction of condition by time (X^2^ (2) = 8.128, *p* = 0.017). Applying the Huynh–Feldt correction, there was no interaction effect of condition and time on sedentary bouts > 60 min (*F* (1.66, 44.76) = 0.720, *p* = 0.468). There was no interaction between the intervention and time on sedentary bouts > 90 min (*F* (2, 54) = 0.095, *p* = 0.910).

### 3.6. Time Spent in Sustained Sedentary Bouts When Working from Home

Significant reductions were found when comparing time spent in sustained SBs > 60 min (60 min: 162.06 ± 18.02 (baseline), 130.79 ± 17.03 (intervention), −31.27 ± 11.91, *F* (1, 27) = 6.887, *p* = 0.014) and >90 min (90 min: 99.13 ± 14.91 (baseline), 71.16 ± 11.45 (intervention), −27.97 ± 9.39, *F* (1, 27) = 8.870, *p* = 0.006) when comparing the participants’ habitual baseline sedentary patterns to the intervention sedentary patterns (Table 3).

There was no main effect of time, when looking at time spent in sustained sedentary bouts > 60 min (*F* (2, 54) = 1.713, *p* = 0.190) or sedentary bouts > 90 min during working hours (*F* (2, 54) = 1.439, *p* = 0.246).

Mauchly’s test indicated that the assumption of sphericity had been violated when comparing the interaction of condition by treatment (X^2^ (2) = 6.504, *p* = 0.039). Applying the Huynh–Feldt correction there was no interaction effect of condition and time when looking at time spent in sustained sedentary bouts > 60 min (*F* (1.729, 46.84) = 0.872, *p* = 0.411). There was no interaction between the intervention and time on time spent in sustained sedentary bouts > 90 min (*F* (2, 54) = 0.424, *p* = 0.657).
ijerph-20-06294-t003_Table 3Table 3This table presents the sedentary behaviour patterns during self-reported occupational hours throughout the baseline (Week 1–3) and intervention (Week 4–6). Presented as Mean ± Standard Error (95% Confidence Intervals).Occupational Sedentary Behaviour PatternsBaselineInterventionMean Change*p**d*Number of SB > 40 (Mins)2.95 ± 0.18 2.79 ± 0.21 −0.16 ± 0.12 0.1930.062(2.56–3.33)(2.36–3.22)(−0.04–0.09)**Time in SB** > **40 (Mins)****226.57 ± 18.13****198.51 ± 18.5****−28.06 ± 10.03****0.009****0.225****(189.37–263.76)****(160.55–236.47)****(−48.65–−7.47)**Number of SB > 60 (Mins)1.64 ± 0.16 1.38 ± 0.16 −0.25 ± 0.13 0.0530.132(1.30–1.97)(1.05–1.72)(−0.51–0.00)**Time in SB** > **60 (Mins)****162.06 ± 18.02****130.79 ± 17.03****−31.27 ± 11.91****0.014****0.203****(125.08–199.03)****(95.85–165.73)****(−55.71–−6.82)****Number of SB** > **90 (Mins)****0.77** ± **0.11**
**0.56** ± **0.08**
**−0.21** ± **0.07**
**0.009****0.228****(0.54–1.00)****(0.39–0.73)****(−0.36–−0.06)****Time in SB** > **90 (Mins)****99.13 ± 14.91****71.16 ± 11.45****−27.97 ± 9.39****0.006****0.247****(68.54–129.73)****(47.67–94.66)****(−47.24–−8.70)**Total Number of SB24.32 ± 2.01 25.17 ± 2.13 +0.84 ± 0.89 0.3520.032(20.21–28.44)(20.79–29.54)(−0.98–2.67)Total Time Spent in SB (Mins)383.71 ± 13.57 366.49 ± 12.91 −17.26 ± 9.61 0.0840.107(355.86–411.56)(339.95–392.94)(−36.98–2.45)Total Sedentary Occupational (h)6.40 ± 0.23 6.11 ± 0.22 −0.29 ± 0.16 0.0840.073(5.93–6.86)(5.67–6.55)(−0.62–0.04)Abbreviations: h; hours. Mins; minutes. SB; sedentary bouts. **Bold** indicates that a significant change was found for the reported outcome variable (*p* < 0.05).


### 3.7. Total Waking Daily Sedentary Time

There was a significant difference between the baseline (11.41 ± 2.82 h) and intervention (10.75 ± 2.70 h) phases (*F* (1, 27) = 16.882, *p* < 0.001) (Table 4).

There was no main effect of time when looking at daily sedentary time (*F* (2, 54) = 1.271, *p* = 0.289). Sphericity was assumed as the Mauchly’s test (X^2^ (2) = 5.353, *p* = 0.069) did not indicate a violation.

There was no interaction between the intervention and time on daily sedentary time (*F* (2,27) = 0.002, *p* = 0.998). Sphericity was assumed as the Mauchly’s test (X^2^ (2) = 0.918, *p* = 0.632) did not indicate a violation.

### 3.8. Time in Bed

There was a significant difference between time in bed between the baseline (8.10 ± 0.16 h) and intervention (8.37 ± 0.15 h) conditions (*F* (1, 27) = 8.208, *p* = 0.008) (Table 4).

There was no main effect of time when looking at time in bed (*F* (2, 54) = 0.407, *p* = 0.668). Sphericity was assumed as the Mauchly’s test (X^2^ (2) = 0.964, *p* = 0.965) did not indicate a violation.

There was no interaction between the intervention and time on daily sedentary time (*F* (2,27) = 0.161, *p* = 0.852). Sphericity was assumed as the Mauchly’s test (X^2^ (2) = 0.782, *p* = 0.676) did not indicate a violation.
ijerph-20-06294-t004_Table 4Table 4This table presents the PBs over the 24 h period during the baseline (Week 1–3) and intervention (Week 4–6). Presented as Mean ± Standard Error (95% Confidence Intervals).Total 24-Hour Physical BehavioursBaselineInterventionMean Change*p**d***Sleep (h)****8.1 ± 0.16****8.37 ± 0.15****+0.26 ± 0.09****0.008****0.233****(7.77–8.43)****(8.06–8.68)****(−0.45–−0.08)****Waking Sedentary (h)****11.41 ± 2.82****10.75 ± 2.70****−0.66 ± 0.16****< 0.001****0.385****(10.83–11.99)****(10.20–11.29)****(−0.99–−0.33)**Standing (h)3.15 ± 0.18 3.34 ± 0.2 +0.19 ± 0.12 0.1080.093(2.77–3.52)(2.92–3.76)(−0.05–0.43)*** Stepping (h)****1.32 ± 0.09****1.42 ± 0.09****+0.10 ± 0.03****0.002****0.295****(1.13–1.52)****(1.24–1.6)****(0.04–0.15)****Non-Wear Time (h)****0.03 ± 0.01****0.14 ± 0.05****+0.11 ± 0.05****0.035****0.154****(0–0.06)****(0.03–0.24)****(0.01–0.21)****LIPA (h)****0.87 ± 0.05****0.94 ± 0.05****+0.07 ± 0.02****0.001****0.324****(0.35–0.56)****(0.83–1.05)****(0.03–0.11)**MVPA (h)0.46 ± 0.05 0.48 ± 0.05 +0.02 ± 0.03 0.4020.026(0.35–0.56)(0.38–0.59)(−0.03–0.08)MVPA (Mins)27.25 ± 2.99 28.85 ± 3.08 +1.601 ± 1.64 0.3360.034(21.13–33.38)(22.54–35.17)(−1.75–4.96)Steps6638.44 ± 538.74 6966.32 ± 507.94 +327.88 ± 182.74 0.0840.107(5533.04–7743.83)(5924.1–8008.53)(−47.08–702.88)Abbreviations: h; hours. Mins; minutes. LIPA; light-intensity physical activity. MVPA; moderate-to-vigorous physical activity. **Bold** indicates that a significant change was found for the reported outcome variable (*p* < 0.05). Time is reported as hours, with minutes expressed as decimals 0–100, whereby any reader interested in converting hours to minutes can multiply the reported outcome variable by 60; for example (1.45 h × 60 = 87 min). * Stepping: The stepping reported is the total accumulated stepping time, where the intensity of the stepping is then further classified into LIPA and MVPA time.


## 4. Discussion

This study investigated the effect of passive prompts on adults in Ireland who held desk-based occupations and were WFH in the Republic of Ireland. Every 45 min, the intervention passively prompted both a break in occupational sitting time and an increase in occupational LIPA, by walking for five minutes. This passive prompt intervention resulted in a significant increase in occupational LIPA (+14.29%) and, furthermore, significantly reduced the number of prolonged occupational SB > 90 min as well as the time spent in sustained SB > 60 and >90 min during their working hours compared to baseline (Table 4). However, whilst a small reduction in the number of prolonged occupational SB > 60 min was seen during the intervention (1.38 ± 0.16) compared to baseline (1.64 ± 0.16), this did not reach significance (*p* = 0.053). Therefore, this study addressed our aims and reports desk-based adults’ PBs and sedentary behaviour patterns when WFH during COVID-19 restrictions in Ireland and the effect of passive prompts on PBs and the pattern of sedentary behaviour accumulation.

The first and third *a priori* alternative hypotheses can be accepted suggesting that passive prompts that recommend both (1) breaking occupational sitting and (2) increasing light-walking were successful in improving occupational sedentary behaviour patterns (both the number of SB and time spent in sustained sitting > 90 min) and LIPA. However, the null hypothesis, for hypothesis two, must be accepted as the number of SB > 60 min did not significantly change (*p* = 0.053), although decreasing, and where the time spent in sustained sitting > 60 min did significantly (*p* = 0.006) reduce when comparing the intervention phase to the baseline phase.

Previous research utilising passive prompts every 60 min compared two groups of participants who received two different messages along with the prompt using an activPAL accelerometer [29]. The first group “Stand” received a prompt stating “Hello, please get out of your chair”, and the second group “Step” received a prompt stating “Hello, please get up and walk at least 100 steps” coupled with a pedometer [29]. The stand group significantly increased stepping time by 5.5 min (14%), whereas the step group significantly increased stepping time by 12 min (29%) compared to their respective baseline measurements. Within this study, we observed a significant 4.2 min (12.96%) increase in stepping time when passively prompting participants with two statements at once (1) “Time for a break!” and (2) “Please complete five minutes of light walking around your room/house/garden/street” every 45 min. This study adds to the current body of literature supporting passive prompts to increase stepping time during occupational hours. The greater change in stepping time observed by Swartz et al. [29] despite smaller step goals (100 steps vs. 5 min of walking) and less frequent breaks (every 60 min vs. 45 min) may relate to the study characteristics and the working environment measured. Our participants may have had limited space to walk when WFH compared to Swartz et al. [29] who were working in an office environment. Furthermore, we evaluated the effects of passive prompts over three working weeks, whereas Swartz et al. [29] measured over three working days; hence, the impact of the passive prompts may be less pronounced when averaged over three weeks as opposed to three days. The participants’ adherence to the passive prompts could have also led to this smaller-than-expected change in stepping, although still a significant effect. Therefore, this study strengthens the current body of literature and provides novel device-based evidence of the effect of passive prompts on occupational PBs and sedentary behaviour patterns over a longer measurement period. Suggesting that passive prompts can sustain behaviour change over three weeks. This is an important finding as sitting is recognised as a strong habit [30]. The habit of sitting is difficult to unlearn and, therefore, raising continuous awareness via passive prompts may be needed [31].

Previous research has investigated occupational sitting and the effect of real-time prompts (triggered by continuous sitting of 30–60 min), via a pressure sensor inserted inside a chair cushion [6]. This real-time prompt software would move from green to amber and then red, and only reset after five minutes of not sitting [6]. This study by Gilson et al. [6] showed a significant reduction in occupational sedentary behaviour by 8% (72-min/day, *p* = 0.018), replacing the sedentary behaviour proportionally with LIPA (8%, *p* = 0.018) and a reduction in total sitting time of −13 min (*p* > 0.05) and reduction in the longest bout by –15 min (111 to 96 min). The study by Gilson et al. [6] is comparable to our own despite Gilson et al. using real-time prompts compared to passive prompts as employed within this study. Within this current study, we observed a 3.37% reduction in occupational sedentary time, which equated to –17.26 ± 9.61 min (*p* = 0.084) compared to baseline.

When looking at the 24 h PBs (See Table 4), we found a significant increase in sleep (time in bed), stepping, and LIPA and a significant decrease in waking sedentary behaviour during the intervention phase compared to baseline. Furthermore, a small but significant increase in non-wear time was detected from baseline to the intervention phase when observing the 24 h PBs (See Table 4). This increase in non-wear time could be a contributing factor to some of the reported decrease in 24 h total sedentary time and possibly due to participant fatigue with the methodology, but the increase in non-wear time is not large enough to substantially contribute to the much larger decrease in total waking sedentary time.

A deeper focus on sedentary behaviour patterns found a significant reduction in the number of SBs > 40, >60, and >90 min and the time spent in sustained SB > 40, >60, and >90 min over the 24 h period (Appendix A). When looking at non-work hours, we found a significant reduction in the number of SB > 60 min (Appendix A). This builds on previous research by Swartz et al. [29] who investigated passive prompts in the workplace, during working hours, but did not assess PBs outside of occupational hours. The current study shows that there were no compensatory changes to PBs outside of the participants’ working hours and that the number of prolonged SB (>60 min), in non-working, hours was significantly reduced when the participants were not passively prompted (Appendix A).

Overall, the findings from this study are positive as it shows that passive prompts during occupational hours can reduce prolonged sustained sitting, which has been recognised as a distinct health risk [19]. Furthermore, although the passive prompt intervention was only carried out during occupational hours, this study observed no negative compensatory changes in PBs during leisure or over the 24 h period, suggesting that the passive prompt intervention during working hours benefited the participants’ overall day. By implementing passive prompts and reducing the habit of prolonged desk-based sitting when working, positive long-term health impacts may be achieved [1]. This study observed a reduction in overall 24 h sedentary time (−32.13 ± 9.35 min, *p* = 0.002) and showed that the time was reallocated between, standing, LIPA, and MVPA, with only the increase in LIPA achieving significance. This does not follow previous isotemporal substitution models, where a direct reallocation from one behaviour to another is generally used to predict a change in health outcomes. Therefore, this study highlights the need for further studies to address the impact of the relatively small changes in daily PBs, with compositional data analysis, where PBs are not kept constant, on indices of cardiometabolic health.

Previous research has shown that the total time spent sedentary independent of exercise has been detrimentally associated with several biomarkers such as insulin, waist circumference, high-density lipoprotein, cholesterol, triglycerides, and C-reactive protein [32,33]. Hence, by reducing total sedentary time via passive prompts, a potential benefit to cardiometabolic biomarkers may have occurred. Further investigation is warranted in a larger sample that assesses cardiometabolic health markers to assess the benefits of reducing total sedentary time and the number of prolonged SBs. This call for future research is further supported by previous research around sedentary breaks. Healy et al. [18,32] showed via a linear regression model that after adjusting for total sedentary time and MVPA time, that sedentary breaks were beneficially associated with fasting plasma glucose, waist circumference, and C-reactive protein. Lastly, this study observed a significant increase in both occupational and 24 h LIPA, and it has been shown that regardless of the domain in which PA is accumulated, the total volume of PA is assumed to be beneficial [34].

This study provides device-based evidence of the accrued time spent sedentary during occupational hours when WFH, with participants who have desk-based occupations during COVID-19. This study found participants were sedentary for 77.71% (6.52 ± 0.23 h (391.2 ± 13.8 min)) of their working hours (8.39 ± 0.15 h) when WFH at baseline. The total occupational sedentary time is comparable, although slightly higher, to one other study, which had a similar sample size of 27 and a thigh-worn accelerometer (Axivity AX3 (Axivity Ltd., Newcastle, UK)), but only measured over a seven-day period. A similar study by Hallman et al. [9] reported occupational sedentary behaviour when WFH of 361 min/day and when working in an office 373 min/day in Sweden during COVID-19 [9]. Whereas another study, which investigated Brazilian office workers’ PBs that did not delineate domains (i.e., work and leisure) but presented whole workdays showed office workers to accumulate 667 ± 85 min sedentary or 46.3% of their whole day when WFH during COVID-19 compared to 689 ± 69 min/day (47.9% of the whole day) pre COVID-19 in Brazil [8]. When we compare total sedentary behaviour during the workday from our population of adults in Ireland who were WFH, they showed at baseline to accumulate 672.56 ± 18.62 min/day, which would be comparable to the findings from Brusaca et al. [8]. Previous office-based research investigating desk-based occupations found employees sat for 5.37 h [35] and 5.55 ± 1.02 h or 71% of working hours [7]. These previous studies and this study demonstrate that individuals WFH still accumulate high levels of sedentary behaviour (6.52 ± 0.23 h or 77.71% of work hours) during their working hours, which reinforces that the habit of sitting is prevalent even when WFH due to the desk-based nature of modern work. Overall, this suggests that total sedentary time during occupational hours is a prominent issue across countries and warrants further research that provides evidence that reducing and/or interrupting this behaviour will provide health benefits.

### 4.1. Strengths and Limitations

#### 4.1.1. Strengths

This study addresses the sparse literature of habitual PBs and sedentary behaviour patterns of individuals with desk-based occupations who are WFH, by providing device-based evidence of the habitual behaviour of this emerging population. Furthermore, this study provides novel evidence surrounding the effect of electronic passive prompts on breaking prolonged bouts of sitting during occupational hours and increasing LIPA.

To the authors’ knowledge, this study is the first to measure PBs over this length of time (six weeks), continuously, following a 24 h wear protocol with a wearable accelerometer. This measurement period allows the authors to be more confident that habitual behaviour was recorded and that the likelihood of measurement reactivity was reduced due to the greater number of measurement days.

A key strength of this study is that due to the study design and 24 h measurement, we could process, analyse, and present the sedentary behaviour patterns during occupational, leisure, and the 24 h period. This allowed the authors to observe and report the effect of the passive prompt intervention in its intended domain (occupational hours) but also shows if any compensatory effects were observed during leisure time and how the intervention influenced the overall day. Finally, our analysis of PBs during occupational hours was based on the participants’ self-reported work hours for each day and, therefore, individualised and reflective of their actual workday, as we did not assume a generic 9 a.m.–5 p.m. workday.

#### 4.1.2. Limitations

A limitation of this study and of passive prompts is that the passive prompts assumed that the sitting patterns of the participants were generic [6] and were triggered regardless of time spent sitting.

Due to the remote nature of this study and the fact that it was carried out during COVID-19, no markers of cardiometabolic health were measured and, therefore, we can only report on the effect of passive prompts on PBs and sedentary behaviour patterns and cannot allude to any potential health benefits directly. Furthermore, as the study was conducted during COVID-19, the findings may not be generalisable to individuals currently WFH now that the restrictions in place to mitigate the transmission of COVID-19 have lifted across Ireland, as individuals may now be leaving the house more often. However, as the baseline PBs are comparable to previous office-based literature pre-COVID-19 [6,9] this could suggest that this evidence base is generalisable to desk-based occupations both office-based and when WFH.

### 4.2. Implications and Future Directions

This study has shown that passive prompts, delivered independently and remotely, were effective in reducing the number of prolonged occupational sedentary bouts > 90 min and increasing LIPA during occupational hours. This has significant implications for future workplace health promotion studies as this study has shown that passive prompts can be used to both interrupt and reduce sedentary time but also increase LIPA, with both of these outcomes associated with potential improvements in cardiometabolic health outcomes.

More research is necessary to provide evidence on the potential health benefits of interrupting prolonged occupational sitting with frequent short bouts of LIPA with individuals who hold desk-based occupations in a free-living setting. Furthermore, future research should investigate complementary strategies to increase desk-based workers’ engagement in following passive prompts and/or increasing incidental LIPA, supporting passive prompts and active breaks with evidenced-based behaviour change theories, such as the TPB.

## 5. Conclusions

Adults in Ireland with desk-based occupations who were WFH during COVID-19 were found to spend a large percentage of their working hours sedentary. The electronic passive prompt intervention was successful in increasing LIPA and reducing the number of prolonged SB (>90 min) during occupational hours and time spent in sustained sitting (>40, >60, and >90 min). Future research could incorporate passive prompts to both interrupt prolonged occupational sitting and increase LIPA whilst investigating the effect on cardiometabolic health markers.

## Figures and Tables

**Figure 1 ijerph-20-06294-f001:**
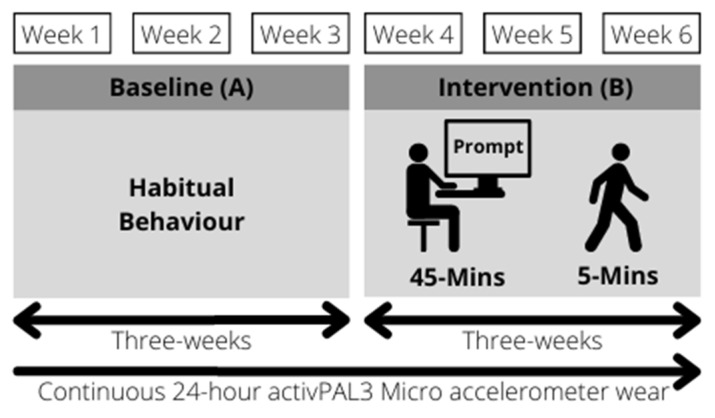
Study design and intervention schematic.

**Figure 2 ijerph-20-06294-f002:**
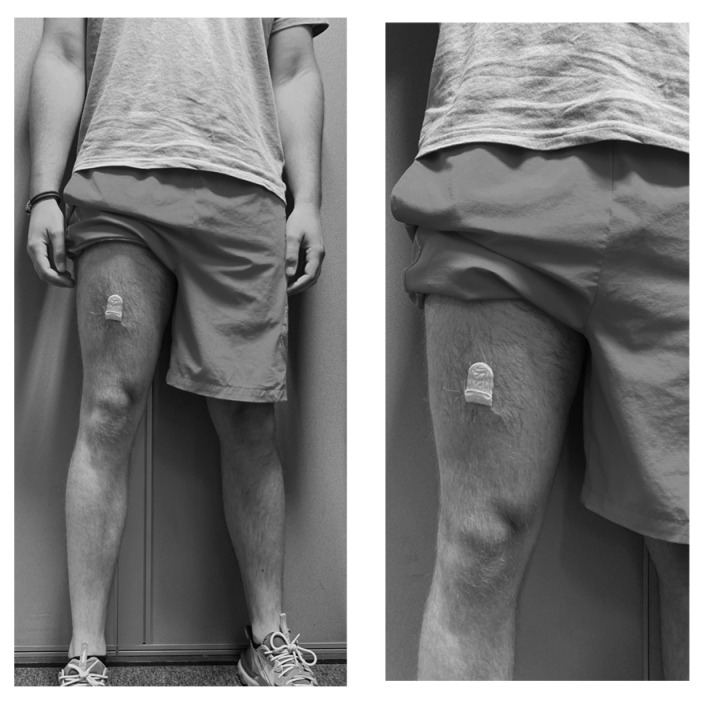
activPAL3 fitting and placement on the participants’ midline of the anterior aspect of their right thigh, using a nitrile sleeve and waterproof Tegaderm.

## Data Availability

The data presented in this study are available on request from the corresponding author. The data are not publicly available due to its use in ongoing PhD research.

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
