# Peer review of "The Effect of an Electronic Passive Prompt Intervention on Prolonged Occupational Sitting and Light-Intensity Physical Activity in Desk-Based Adults Working from Home during COVID-19 in Ireland"

_ijerph, 2023, doi:10.3390/ijerph20136294_

Round 1
Reviewer 1 Report
Thank you for study. The number of subjects in the article is quite small, but I think it will be useful to evaluate e Effect of an Electronic Passive Prompt Intervention on Pro- 2 longed Occupational Sitting and Light-Intensity Physical Ac- 3tivity in Desk-Based Adults Working from Home During 4 COVID-19 in Ireland.
There are a few points that should be added in the Method section.
The effect of shoes during walking is quite high. In the study, no explanation was written about the shoes that individuals wear during walking, I think it should be added.
Inclusion and exclusion criteria can be clearly explained. Cardiovascular, orthopedic or neurological problems of the evaluated persons should be stated.
Author Response
Please see the attachment with author comments and point-by-point feedback. The authors would like to thank the Reviewer for their time and expertise and strengthening the overall manuscript.

Reviewer 2 Report
The article describes an interesting experiment involving the use of passive prompts to modify occupational physical behaviors (PBs) and bouts of prolonged sitting among desk-based workers. Experimental design and implementation are correct and results are well documented. It has been shown that during interventions, the number of prolonged occupational sedentary bouts >90-minutes were reduced compared to baseline. The time spent in sustained sitting >60 and >90-minutes when compared to baseline sedentary patterns also decreased. However, light-intensity physical activity (LIPA) significantly increased during the intervention.
However, the authors did not pay attention to the low interest in participating in the intervention among desk-based workers. The invitation to participate was addressed through mass emailing to staff and affiliates of the University of Limerick via an email distribution list and through a public recruiting call for participants on Twitter. I miss comparing the number of people the invitation could have reached with the number of 60 individuals expressed interest in participating in this intervention. I believe that in addition to further research to provide evidence on the potential health benefits of interrupting prolonged occupational sitting with frequent short bouts of LIPA, effective ways to increase the participation of desk-based workers in such activities should be analyzed and developed.
Author Response

(The authors gave the same response as above.)

Reviewer 3 Report
Dear, Authors,
Thank you for your work! That was a pleasure to read it. If there were some tiny mistakes, unfortunately, I didn't see them. Good luck!

Author Response
Please see the attachment with author comments and point-by-point feedback. The authors would like to thank the Reviewer for their time and expertise.

Reviewer 4 Report
This is a quasi-experimental study (or pre-post design) evaluating the effects of a computer prompt in desk-based workers in light PA (LIPA) time and sedentary time. The intervention had the potential of adding ~50 min of LIPA [hopefully reducing 50 min of sitting (sedentary behavior)] while working. It is a very well written manuscript, with enough detailed information generally to warrant replication, and despite the limitations of a pre-post design in assessing cause-and-effect relationships, the small sample size, a particular context (Covid-19 pandemic time), small period of intervention, it has its merits. I’m struggling, however, to understand the ANOVA model authors have used to compare pre-post physical behaviors (and during the intervention maybe? Differences between wks 4,5 and 6, not wks 1, 2 and 3?), and to read the descriptive statistics of most of the time variables as the decimals are in 0-100 scale not in hours and minutes scale (i.e., 0–60).
Specific comments:
Lines 45–56: Sedentary behavior may be due to tasks not necessarily the surrounding environment. Can you framework this research into a physical activity theory(ies)? It would be great.
Has this trial been registered?
Line 109: What about the disadvantages? For weighing pros and cons…
Line 110–128: Please report sample size determination.
Line 150: add “triaxial”.
Line 223–224: But is it comparable? It might be difficult to interpret your results and compare with other studies…
Lines 269–270: I suggest reporting this in Results and reporting how the decision was made. Have you used the date of “lying” stated in lines 222–223?
Lines 279–281: I suggest that in the section 2.2., it should be included that individuals working hybrid were eligible as long as… [provide the reason(s), as it could be replicated]
Lines 283–302: I’m a little bit confused with your ANOVA model. You state that you have 2 sources of within-subject variation, however they are both time-dependent variables. Is Baseline variable the mean of the valid days of the 3 pre-intervention weeks, or just the last week (wk 3)? If so, what represents your X 3 (Time: Week 1/Week 2/Week 3)? You wanted to compare whether during wk 4, 5, and 6 physical behaviors were different? Information regarding what are Baseline and Intervention variables (composite scores???) should be detailed otherwise is very hard to understand. You may also consider conducting separate analyzes (Baseline and Intervention) and for differences across weeks (if I understood your goals).
Line 303: Please, state how to interpret Cohen’s d values.
Results:
How many were in the hybrid situation?
Table 1: Can the authors report BMI?
I’m usually not very demanding with the decimals of time variables, but because this research is all about time, I suggest presenting the decimals of hours in minutes (0-60), not the decimals 0–100. Some values may be easy to convert mentally (e.g., 0.5 = half an hour or 30’) but other may be more tricky or even misleading, e.g., 0.05 hours represents 3 minutes, not 5 minutes. [I close my eyes regarding years (decimals in months)]
Occupational LIPA is your main outcome measure however it didn’t deserve any subheading highlight!
Table 2. Typos on the intersections Work (h) X Mean Change [(-0.45--0.07)] and LIPA (h) X Baseline [(0.30 ± 0.40)].
Lines 345–346: Here again the descriptive statistics look strange. You have the counts or frequencies of how many people spend more than 60 mins in sedentary behavior. Maybe just one decimal 1.6 ± 0.2 (1.6 times [±0.2] people stay sedentary for more than 60 minutes is more intelligible for most readers)
Line 410: Calculating the values pre-post in Table 2 and 4, I can’t reach this (14.29%) LIPA percent change.
Lines 567–569: Over a 3-week period (intervention) was it expected meaningful changes in cardiometabolic health?
Author Response

(The authors gave the same response as above.)

Round 2
Reviewer 4 Report
I'm very satisfied with the explanations and amendments authors' have provided. I wonder, nonetheless, whether comparing PB throughout Intervention weeks wouldn't be more informative about, say regression to the mean or regression to the baseline phenomenon, or in other words, if the electronic passive prompt naturally falls in efficacy from wk 4 to wk 6 due to "users passive prompt fatigue". Or even if it increases or maintains its initial increments due to a potentially sustained “behavioral change”.
In tables you may use a dash – instead of the minus - symbol to separate the min–max values.